# Glial Cell Dysfunction in *C9orf72*-Related Amyotrophic Lateral Sclerosis and Frontotemporal Dementia

**DOI:** 10.3390/cells10020249

**Published:** 2021-01-28

**Authors:** Mehdi Ghasemi, Kiandokht Keyhanian, Catherine Douthwright

**Affiliations:** Department of Neurology, University of Massachusetts Medical School, Worcester, MA 01655, USA; kiandokht.keyhanian@umassmemorial.org (K.K.); catherine.douthwright@umassmed.edu (C.D.)

**Keywords:** *C9orf72* gene, *C9orf72* repeat expansion mutation, amyotrophic lateral sclerosis (ALS), frontotemporal dementia (FTD), astrocytes, microglia, glial cells

## Abstract

Since the discovery of the chromosome 9 open reading frame 72 (*C9orf72*) repeat expansion mutation in 2011 as the most common genetic abnormality in amyotrophic lateral sclerosis (ALS, also known as Lou Gehrig’s disease) and frontotemporal dementia (FTD), progress in understanding the signaling pathways related to this mutation can only be described as intriguing. Two major theories have been suggested—(i) loss of function or haploinsufficiency and (ii) toxic gain of function from either *C9orf72* repeat RNA or dipeptide repeat proteins (DPRs) generated from repeat-associated non-ATG (RAN) translation. Each theory has provided various signaling pathways that potentially participate in the disease progression. Dysregulation of the immune system, particularly glial cell dysfunction (mainly microglia and astrocytes), is demonstrated to play a pivotal role in both loss and gain of function theories of *C9orf72* pathogenesis. In this review, we discuss the pathogenic roles of glial cells in *C9orf72* ALS/FTD as evidenced by pre-clinical and clinical studies showing the presence of gliosis in *C9orf72* ALS/FTD, pathologic hallmarks in glial cells, including TAR DNA-binding protein 43 (TDP-43) and p62 aggregates, and toxicity of *C9orf72* glial cells. A better understanding of these pathways can provide new insights into the development of therapies targeting glial cell abnormalities in *C9orf72* ALS/FTD.

## 1. Introduction

Amyotrophic lateral sclerosis (ALS, also known as Lou Gehrig’s disease) and frontotemporal dementia (FTD) are two devastating neurodegenerative diseases with a high burden on society. It is currently believed that ALS and FTD are parts of a disease spectrum that share clinical, genetic, and pathological findings. Clinically, 30–50% of ALS patients have cognitive deficits, and ~15% of patients with FTD exhibit symptoms/signs of ALS [1,2]. Histopathological studies have also shown that >97% of ALS and ~50% of FTD patients aggregate inclusions of the TAR DNA-binding protein 43 (TDP-43) in both affected neurons and glial cells [3,4,5,6,7]. In 2011, a trio of teams [8,9,10] discovered a GGGGCC (G_4_C_2_) nucleotide repeat expansion mutation in the first intron of the chromosome 9 open reading frame 72 (*C9orf72*) gene as the most frequent genetic cause in up to 35–45% of familial ALS, 5–20% of sporadic ALS, 15–25% of familial FTD, and 6–7% of sporadic FTD patients [11]. Although the number of hexanucleotide repeats varies considerably among these patients, overall, repeat numbers of <30 are considered to be non-pathogenic [12]. The relationship between repeat expansion size and phenotype is still equivocal, which could be due to somatic variability in expansion size [2]. *C9orf72* ALS patients have a mean onset age of 57 years old and a median survival rate of 30–37 months [13]. Although more frequent bulbar onset has been reported in *C9orf72* ALS patients compared to ALS patients without a *C9orf72* repeat expansion mutation [14,15,16], this is still debatable, because a recent multi-center prospective natural history study on *C9orf72* ALS cases reported a higher rate of limb (54%) than bulbar (39%) onset [13]. What is undebatable, however, is that the prevalence of FTD is significantly higher in *C9orf72* ALS cases, accompanied by higher rates of disease progression and prominent cognitive/behavioral changes [14,17] (Figure 1). Co-morbid dementia is present in 50% of *C9orf72* ALS patients [14]. *C9orf72* FTD patients have also more common psychotic features and irrational behavior compared to non-*C9orf72* FTD cases [18,19].

Although much more research is needed to understand the normal function of *C9orf72* in humans, the discovery of *C9orf72* repeat expansion mutations as the most common genetic etiology in ALS/FTD has opened a new avenue of research for elucidating disease mechanisms and, ultimately, therapeutic approaches for this fatal disease. Initial observations identified decreased levels of C9orf72 protein in several brain/spinal cord regions [8,10,20,21,22,23,24,25], suggesting a loss of function or haploinsufficiency as a main pathogenic mechanism. Using a variety of *C9orf72* knock out/down animal models, several mechanisms related to this theory were suggested, including aberrant autophagy, disrupted endosomal/lysosomal or endoplasmic reticulum (ER)-Golgi transport systems, and excitotoxicity [2,26]. Although this hypothesis still explains several aspects of *C9orf72* ALS/FTD pathogenesis, other investigators have proposed a gain of toxic function, through the generation of toxic RNA repeats and dipeptide repeat proteins (DPRs) [2]. In recent years, compelling evidence indicates a role for immune dysregulation, particularly related to glial cell abnormalities, as an important mechanism underlying *C9orf72* ALS/FTD pathogenesis. Here, we comprehensively review the current literature on the pathogenic roles of glial cells, focusing on microglia and astrocytes, in *C9orf72* ALS/FTD as evidenced by pre-clinical and clinical studies.

## 2. Overview of Pathogenic Mechanisms Underlying *C9orf72* Repeat Expansion Mutation in ALS/FTD

### 2.1. Loss of Function Mechanisms

The *C9orf72* gene consists of 11 exons (including two alternate non-coding first exons—1a and 1b) [8]. Through alternative splicing, it can be transcribed into three transcript variants (Figure 2). The (G_4_C_2_)_n_ repeat expansion mutation is located in intron 1 of variants 1 and 3, whereas in variant 2, it is located within the promoter region (Figure 2). Therefore, the repeat expansions are not incorporated into variant 2 pre-mRNA. Two protein isoforms are encoded from these transcript variants—(i) a short 222-amino acid protein (24 KDa) from variant 1 and (ii) a long 481-amino acid protein (54 KDa) from variants 2 and 3 [8,9]. Compared to variants 1 and 3, expression of variant 2 is higher in the central nervous system (CNS) relative to other tissues [27,28], especially in the fetal brain and adult cerebellum and frontal cortex, and has lower expression in the hippocampus [8]. Based on immunohistochemical studies, the C9orf72 protein is mainly a neuronal cytoplasmic protein, localizing largely at the presynaptic terminals [29]. More investigation using specific antibodies for either short or long C9orf72 protein has also demonstrated that long C9orf72 protein has a diffuse cytoplasmic presence in neurons with a large antibody staining in cerebellar Purkinje cells [30]. However, the short C9orf72 protein has a very specific nuclear membrane localization in healthy neurons, with evident plasma membrane relocalization in the ALS motor neurons [30]. Cellular expression and localization of C9orf72 protein isoforms also alter throughout the development [31].

Investigations on carriers of *C9orf72* expansions have found reduced levels of *C9orf72* transcript variants (particularly variants 1 and 2) in the frontal cortex [8,10,20,21,22,23,24], cerebellum [22,23,24,25,32], motor cortex [25], cervical spinal cord [25], induced pluripotent stem cell (iPSC)-derived neurons [25,27,33,34], and blood lymphocytes [8,20,35]. Higher levels of variant 1 were linked with prolonged survival after disease onset in expansion carriers [24]. This could be an important consideration for the development of new therapeutic approaches targeting *C9orf72*. Moreover, C9orf72 protein levels may be reduced in the frontal cortex in these subjects [23,30]. The above findings led to the initial assumption that the loss of C9orf72 protein level or function may be involved in the disease pathogenesis. Accordingly, several mechanisms have been proposed, as we discuss below.

Initial investigations revealed that C9orf72 protein shows structural homology to the differentially expressed in normal and neoplastic cells (DENN) guanine nucleotide exchange factor (GEF) proteins [36]. Functioning as a GEF, the DENN domain of C9orf72 protein is predicted to interact with Rab GTPases [36,37,38,39], which play crucial roles in both vesicular trafficking and autophagy. Immunohistochemistry of *C9orf72* ALS patient motor neurons shows enhanced colocalization between C9orf72, Rab7, and Rab11 (involved in late endosome maturation or endosome recycling, respectively) compared with controls [40]. Additionally, decreased expression of *C9orf72* was shown to potentiate the aggregation and noxiousness of Ataxin-2 with intermediate-length polyglutamine expansions (Ataxin-2 Q30x) but not of Ataxin-2 with normal polyQ length (Ataxin-2 Q22x). Notably, Ataxin-2 Q30x is a genetic modifier of ALS/FTD [41,42,43,44]. Sellier et al. (2016) showed that depletion of *C9orf72* partially deteriorated neuronal survival and synergized with Ataxin-2 Q30x toxicity to cause motor neuron degeneration, proposing a double-hit pathological contribution to ALS/FTD [45]. Consistent with the above findings, autophagy initiation was found to be disrupted in *C9orf72*-knockdown human cell lines or primary neurons [38,45], causing aggregation of cytoplasmic p62 and TDP-43 [38,45], both of which are histopathological characteristics of ALS/FTD. Ultimately, these findings indicate a potential disruption in autophagy as a loss-of-function mechanism for *C9orf72* ALS/FTD disease pathogenesis (Figure 3).

Another potential loss-of-function mechanism is that of disrupted lysosomal degradation. Reduced endocytosis and impaired endosomal/lysosomal trafficking have been demonstrated in *C9orf72* knockdown cell lines [40], bone marrow-derived macrophage and microglia from *C9orf72*^−/−^ mice [46], and *C9orf72* ALS patient-derived fibroblasts and neurons [47]. C9orf72 protein has been shown to be localized primarily to early endosomes in iPSC-derived motor neurons [34,48]. Accordingly, fewer lysosomes and reduced vesicular trafficking are observed in iPSC-derived motor neurons from *C9orf72* ALS patients [34]. Mannose-6-phosphate receptors (M6PRs) are a group of transmembrane glycoproteins that target lysosomal enzymes to lysosomes. It has been shown that these receptors are affected by *C9orf72* mutations [34] because they cause clustering of these receptors, slowing their movement [34], and their intracytoplasmic mislocalization (rather than normal perinuclear localization) in *C9orf72* ALS/FTD fibroblasts [47]. Therefore, these changes related to *C9orf72* mutation disrupt lysosomal degradation. Accumulating evidence also indicates that *C9orf72* repeat expansion mutation may negatively affect the ER-Golgi transport system [49,50]. *C9orf72* knockdown impairs endocytic trafficking from the plasma membrane to the Golgi [40,47] (Figure 3).

Neuronal hyperexcitability and related excitotoxicity secondary to aberrant glutamatergic transmission have been suggested as the underlying mechanisms for ALS/FTD pathogenesis [51]. Regulated glutamatergic transmission is a complex process, depending on extracellular glutamate levels, reuptake, and re-synthesis, in addition to activation of postsynaptic glutamate receptors (including *N*-methyl-D-aspartate (NMDA) and non-NMDA such as α-amino-3-hydroxy-5-methyl-4-isoxazolepropionic acid (AMPA) receptors) and related intracellular calcium overload. Accumulating evidence has suggested that this system is involved in the pathogenesis of ALS. However, the only modest effect of the glutamate release inhibitor, riluzole, on the survival of ALS patients indicates that this pathway is not the sole mechanism for ALS pathogenesis. Nevertheless, in recent years, investigators have tried to elaborate a link between ALS/FTD gene mutations, including *C9orf72* repeat expansions, and neuronal hyperexcitability/excitotoxicity mechanisms. The expression of kainate receptors and voltage-gated Ca^2+^ channels in iPSC-derived motor neurons, cell surface levels of the NMDA receptor GluN1 and the AMPA receptor GluR1 on neurites, and dendritic spines of iMNs from *C9orf72* ALS/FTD patients are found to be markedly elevated compared to controls [34,52,53]. Glutamate receptors also accumulate at postsynaptic densities in these neurons [34]. Additionally, the post-mortem anterior horn of cervical spinal cord sections from *C9orf72* ALS patients have increased GluR1 expression [53]. High levels of glutamate receptors can lead to hyperexcitability and cell death as a result of glutamate hyperactivation. Accordingly, activation of Kv7 potassium channels was found to improve the survival of *C9orf72* patient-derived and *C9orf72*-deficient iMNs [34]. More recent studies demonstrated that iPSC-derived motor neurons with *C9orf72* mutation had elevated Ca^2+^-permeable AMPAR expression and selective motor neuron susceptibility to excitotoxicity [53], which was eliminated by CRISPR/Cas9-mediated correction of the *C9orf72* mutation in these neurons [53]. Other investigators [25] also found that the *C9orf72* repeat expansion mutation causes nuclear RNA foci sequestering of the enzyme adenosine deaminase acting on RNA 2 (ADAR2), which catalyzes GluR2 editing, linking the *C9orf72* mutation to excitotoxicity. Given the important role of ADAR2 in double-stranded RNA editing, mislocalization of ADAR2, as shown in a recent study on transgenic (G_4_C_2_)_149_ mice, can have detrimental effects on RNA editing [54]. Consistently, ADAR2 knockdown in mice motor neurons slows the rate of degeneration and reduces the loss of neuromuscular synapses in these cells [55]. Agents such as an anticoagulation-deficient form of activated protein C (3K3A-APC), which can lower glutamate receptor levels, are able to decrease excitotoxicity and rescue proteostasis in vivo in both *C9orf72* gain- and loss-of-function mouse models [56].

### 2.2. Challenges in Loss-of-Function Theory

Although based on the above-mentioned studies it was originally proposed that loss of function is the main mechanism underlying the pathology of *C9orf72* repeat expansion mutation in ALS/FTD [34], subsequent studies have challenged this hypothesis. An important initial observation was that neural-specific ablation of *C9orf72* in conditional *C9orf72^−/−^* mice did not induce motor neuron degeneration, defects in motor function, or alter survival [57]. Moreover, several studies have shown that ubiquitous [46,58,59,60,61,62] or CRISPR/Cas9-mediated [61,63] *C9orf72* knockouts throughout development resulted in dysregulation of the immune system in homozygous mice. These mice exhibited a variety of manifestations, including significant changes in myeloid and/or lymphoid cell populations in lymph nodes and spleen, higher levels of inflammatory cytokines, cervical/systemic lymphadenopathy, splenomegaly, glumerulonephropathy, decreased body weight, malignancies, and elevated titers of autoimmune antibodies. This severe phenotype, however, was not observed in haploinsufficiency models of *C9orf72* [64]. Next, studies surveyed the *C9orf72* locus using cap analysis of gene expression sequence data (CAGEseq) and found high gene expression in CD^14+^ monocytes, important cells in innate and adaptive immunity [28]. Taken the above challenges together with the role of *C9orf72* in signaling pathways previously implicated in ALS/FTD [64,65], *C9orf72* haploinsufficiency combined with gain-of-function mechanisms and/or mutations in other modifier genes (for instance, Ataxin-2 Q30x, as discussed above) are possible mechanisms underlying the ALS/FTD pathogenesis.

### 2.3. Gain-of-Function Mechanisms

The dominant inheritance pattern of *C9orf72* ALS/FTD, the nonappearance of ALS or FTD patients with missense mutations or null alleles in the *C9orf72* gene, and the absence of a neurodegeneration phenotype in most of *C9orf72*^−/−^ mice (as discussed above) have argued against the loss of *C9orf72* function theory as the single mechanism of the disease. In fact, more recent evidence points to the gain of toxic functions as the major mechanism underlying neurodegeneration in *C9orf72* ALS/FTD (Figure 3). Accordingly, the adeno-associated virus (AAV)-mediated delivery of a construct that expresses G_4_C_2_ repeats can cause neurodegeneration in mice brain [66]. Several mechanisms have been proposed to explain the toxic gain of function in *C9orf72* ALS/FTD.

When the *C9orf72* repeat expansion mutation was initially identified in *C9orf72* ALS/FTD patients, it was also found that widespread intranuclear RNA foci containing the G_4_C_2_ repeats accumulate in both the brain and spinal cord of these patients [8]. This observation provided a second possible disease mechanism involving a toxic gain of function by repeat-containing RNA. It was also shown that *C9orf72* could be bidirectionally transcribed to the sense G_4_C_2_ or antisense C_4_G_2_ RNA transcript, which can sequester as RNA foci in the affected cells [67,68,69]. These RNA foci were identified in fibroblasts [25,33,70] and motor neurons derived from fibroblast-derived iPSCs from *C9orf72* ALS patients [25,33,70,71]. A similar mechanism was previously suggested in other neuromuscular disorders, including myotonic dystrophy type I, myotonic dystrophy type II, fragile X-associated tremor and ataxia syndrome, and some types of spinocerebellar ataxia, all of which are due to the expansion of nucleotide repeats in non-coding regions [72]. The exact mechanisms by which intranuclear RNA foci cause neurotoxicity/degeneration in ASL/FTD is not completely understood; however, the following mechanisms have been suggested:Binding of RNA foci to RNA-binding proteins (RBPs), forming neurotoxic aggregates [26];Formation of G-quadruplex and R-loop structures [73,74,75,76,77,78,79,80], causing nucleolar stress [77], genomic instability, and an increased DNA double-stranded break [81,82];Formation of other secondary structures such as hairpins [76,77], RNA duplexes, and i-motifs and DNA‒RNA heteroduplexes [83,84,85], which may be toxic for neuronal cells.

One of the most intriguing pathologic models that has been recently proposed as the primary pathology for *C9orf72* ALS/FTD is the formation of dipeptide repeat proteins (DPRs). It was shown that the repeat-containing *C9orf72* transcripts can escape the nucleus and be attached by ribosomal complexes, thereby boosting repeat-associated non-ATG-dependent (RAN) translation that leads to toxic aggregation of polydipeptides or DPRs [77] (Figure 3). The expanded domains in *C9orf72* can undergo RAN translation in all six possible reading frames and across both sense and anti-sense RNA; this results in the generation of five different DPRs (Figure 3) [86]. Overexpression of each DPRs in various cell models [69,87,88,89,90,91,92,93], zebrafish [94,95,96], *Drosophila* [87,97,98], and mice [99,100] have resulted in neurotoxicity and revealed the involvement of several downstream pathways. Among the DPRs, poly-glycine-arginine (poly-GR) and poly-proline-arginine (poly-PR) were the most neurotoxic, and poly-glycine-alanine (poly-GA) exerted less toxicity [20,87,88,89,92,101,102,103]. Other DPRs, such as poly-proline-alanine (poly-PA) and poly-glycine-proline (poly-GP), had less or no toxicity [87,98,102]. Administration of synthetic poly-PR and poly-GR into cultured human astrocytes [104] and poly-GA and poly-GR into primary neurons [105] also caused cellular toxicity. It was further shown that poly-(GA)_15_ fibrillates rapidly and eventually forms toxic flat, ribbon-type fibrils, as demonstrated by transmission electron microscopy and atomic force microscopy [93].

Given the toxic nature of DPRs, these polydipeptides probably affect a variety of downstream pathways that eventually lead to neuronal cell death, which include the following (Figure 3):Impairment of liquid–liquid phase separation (LLPS) through interaction with low complexity domain (LCD) proteins in nucleoli and stress granules [97,106,107,108]. LLPS of key protein and nucleic acid scaffolds play an important role in the biogenesis of diverse membrane-less organelles (e.g., P granules and stress granules in the cytoplasm and nucleoli and paraspeckles in the nucleus) that are essential organizers of subcellular biochemistry, controlling the information processing from genotype to phenotype [109];Binding with and thereby inhibiting translation initiation and elongation factors, causing neurotoxicity [89,110,111,112,113,114];Impairment of ribosomal RNA maturation and abnormal splicing. A pioneering study by Kwon et al. (2014) indicated that exogenous administration of synthetic poly-(GR)_20_ and poly-(PR)_20_ to human astrocytes led to their accumulation in the nucleus and binding to the LCD of hnRNPA2, causing aberrant pre-mRNA splicing and impaired rRNA biogenesis [104];Mitochondrial dysfunction [115,116];Binding with nuclear pore complex proteins, causing a defect in nucleocytoplasmic trafficking [25,30,104,117,118,119], and thereby neurotoxicity.

## 3. Neuroinflammation in *C9orf72* ALS/FTD: Glial Cells Dysfunction

As previously described, the *C9orf72* gene can be expressed in different cell types other than motor neurons and most significantly in the immune system [28,61]. Neuroinflammation is an extremely complex process involving glial cells. Despite its complexity, it is a well-orchestrated symphony of cross talks between different cell types via cytokines and other molecules. Disturbance of this highly evolved function may lead to detrimental effects on the nervous system function [120]. Here, we will discuss the role of glial cells, focusing on microglia and astrocytes and chronic neuroinflammation in the *C9orf72*-related ALS/FTD.

### 3.1. Glial Cells in the Central Nervous System

Although the CNS is traditionally considered an “immunologically privileged site” due to the blood-brain barrier, immunological reactions are still occurring within the CNS by different mechanisms despite the absence of leukocytes and antibodies. CNS homeostasis mainly is regulated by innate immunity [121]. The key mediators of immune reactions within the CNS are glial cells, which are the most abundant cell type in the CNS. Glial cells consist of microglia, astrocytes, and oligodendrocytes [121]. Here, we mainly focus on microglia and astrocytes. Microglia are the resident mono-phagocytic cells in the CNS [122]. During embryogenesis will spread in the brain after being derived from myeloid precursor cells and later would make up to 12% of adult CNS cells [123]. Microglia are generally the first cell type to get activated in response to insults and they are the most motile cell types in CNS [120]. Activated microglia have various physiologic functions including cellular maintenance, innate immunity, the release of trophic and anti-inflammatory factors, and expediting stem cell migration to the site of injury or inflammation [124,125,126]. Microglia morphology alters in response to certain stimuli (e.g., brain injury or immunological stimuli) from resting ramified microglia to an amoeboid form that presents an upregulated series of surface molecules, receptors, and new intracellular proteins/enzymes such as inducible nitric oxide synthase (NOS) and cyclo-oxygenase 2 [121,125]. Astrocytes, on the other hand, are the most abundant glial cells in the CNS. They play variable vital roles, including but not limited to balancing key elements in ionic homeostasis, buffering the action of neurotransmitters (particularly excitatory ones), and secreting growth factors and nutrients [127]. They also contribute to regulating blood-brain barrier function, synaptic plasticity, and neuroprotection [128,129]. Astrocytes generally produce multiple extensions from their cell body and make endfeet at their extended end. Astrocyte’s extensions interact with other cells, including blood vessel’s endothelial cells and pericytes, helping to build the blood-brain barrier [128,130]. Activated astrocytes release trophic factors to help neuronal survival in response to injury. Reactive astrocytes can be divided into A1 and A2 type phenotypically, parallel to what is known for reactive macrophages categorized as M1 and M2. Gene transcriptome analyses demonstrated that A1 astrocytes express inflammatory cytokines and trigger cascades that are harmful and destructive to synapses, while A2 astrocytes express neurotrophic factors and help synapse repair [131]. It seems that different types of injuries may determine which kind of reactive astrocytes would dominate the response. For example, ischemic injuries to CNS provoke an A2 response, but inflammatory insults will trigger A1 reactive astrocytes [131,132]. Generally, under chronic stress conditions such as progressive neurodegeneration, both microglia and astrocytes remain activated, which leads to detrimental outcomes on neuronal cell function due to excess production of different neurotoxic cytokines (e.g., interleukin [IL]-1β and tumor necrosis factor [TNF]-α) and noxious molecules (e.g., excess NO and superoxide anions) (Figure 4) [121].

### 3.2. A Role for Glial Cells in C9orf72 ALS/FTD

Although several studies investigating ubiquitous [46,58,59,60,61,62] or CRISPR/Cas9-mediated [61,63] *C9orf72* knockout in homozygous mice throughout development have inconsistently reported on motor neurons’ involvement in these animals, one unequivocal and yet crucial result has been found—a dysregulation of the immune system exists in homozygous mice (Table 1). This is evidenced by altered myeloid/lymphoid cell populations in lymph nodes/spleen, elevated inflammatory cytokines/autoimmune antibodies, cervical/systemic lymphadenopathy, splenomegaly, and malignancies. High *C9orf72* gene expression was also found in CD^14+^ monocytes [28]. These observations support that (i) complete loss of *C9orf72* results in a systemic pro-inflammatory state possibly driven by myeloid cells in the spleen and lymph nodes and (ii) haploinsufficiency may be enough to affect myeloid cell function and systemic immunity in mice [133]. Additionally, *C9orf72*^−/−^ mice exhibit age-related neuroinflammation [67].

Given the fact that neuroinflammation (e.g., glial activation) has been indicated in the pathophysiology of a variety of neurodegenerative diseases [121], and even though there is still debate whether it is a cause or a consequence of these diseases, the above findings have raised an important question: does immune system dysregulation, and in particular glial cells dysfunction, contribute to the *C9orf72* ALS/FTD pathogenesis?

#### 3.2.1. Evidence for Gliosis in C9orf72 ALS/FTD

To address the involvement of glial cells in the *C9orf72* ALS/FTD pathogenesis, a wide range of studies with different methodology and outcome measurements has been conducted (Table 2). For instance, RNA expression analyses have found that signaling pathways related to interferon (IFN)-γ, IL-1β, complements, and antigen presentation system are markedly upregulated in the brains of *C9orf72* ALS patients compared to sporadic ALS cases [60,136]. Increased glial fibrillary acidic protein (GFAP) expression, an indicator of astrogliosis, has been detected in the frontal cortex and cerebrospinal fluid (CSF) of FTD patients [137,138] and *C9orf72* (G_4_C_2_)_500_ BAC transgenic mice [139]. Moreover, astrocytes exert higher glucose metabolism in *C9orf72* ALS patients compared to other ALS cases [140]. However, *C9orf72^−/−^* mice do not exert any elevated GFAP immunoreactivity [57], suggesting that absent *C9orf72* does not lead to astrogliosis.

Microglial hyperactivation is also found in *C9orf72* ALS/FTD post-mortem brain regions, as evidenced by altered cell morphology and increased markers of gliosis (ionized calcium-binding adaptor molecule 1 (Iba1) and cluster of differentiation 68 (CD68) immunoreactivity), compared to sporadic ALS/FTD cases [4,141,142] (Table 2). CD68 is a lysosomal membrane glycoprotein in mononuclear phagocyte lineage cells, such as macrophages, microglia, osteoclasts, and myeloid dendritic cells [144]. It is widely used as an indicator of phagocytic activity of microglia [145,146], although it can also be expressed on resting microglia [147]. Iba1 is a 17kDa protein that is expressed in all subtypes of microglia, contributing to microglia motility and phagocytosis, and it is associated with microglial activity [148,149,150,151].

Activated microglia were also found to have enlarged lysosomes compared to sporadic ALS cases [60]; the pathologic aspects of this finding are yet to be identified. Further studies are still needed to delineate the relation of gliosis to different stages of the disease process, given the fact that either astro- or micro-gliosis could act as both pro- and anti-inflammatory. Accordingly, an interesting study [100] showed that both elevated CD68 and Iba1 expression are detected at six months of age in the spinal cord of transgenic mice that express poly-(GA)_149_ specifically in neurons [100]. Notably, glial hyperactivation was not present in other brain regions where the neurons did not have poly-GA pathology, and at the time, there was no significant detectable neuronal loss [100]. Notably, at this point, these mice exhibited an enhanced TDP-43 phosphorylation without translocation/inclusions or mild behavioral deficits [100]. When mice were evaluated at one month of age, on the other hand, they did not have the elevation of Iba1 but already had an elevated CD68 expression [100]. These findings indicated two important points—(i) microglial activation might precede severe neuronal dysfunction and (ii) enhanced microglial CD68 expression may precede increased Iba1 expression. Another study showed that six-month-old poly-(GA)_50_ mice did not have elevated Iba1 levels or TDP-43 pathology, but they had behavioral impairments and neurodegeneration [99]. A more recent study also showed that 1.5-month-old poly-(GR)_100_ mice had elevated cortical/hippocampal Iba1 expression, brain atrophy, and neuronal loss, without TDP-43 pathology [114]. The results of these studies may indicate that, firstly, neuron-glia communication might participate in the activation of microglia and, secondly, length or type of DPR and additional factors (e.g., TDP-43 phosphorylation) might differentially control microglial activation in the *C9orf72* pathology.

#### 3.2.2. C9orf72 Pathologic Hallmarks in Glial Cells

Another important issue in the study of glial cells in *C9orf72* pathology is to answer whether these cells also exhibit characteristic *C9orf72* pathology (i.e., RNA foci or DPRs) similar to the affected neurons. Notably, studies investigating the presence of RNA foci or DPRs in glial cells of *C9orf72* ALS/FTD cases or in animal models have consistently demonstrated either no or low levels of RNA foci [67,70,71,152,153] and DPRs [67,86,154,155] in different CNS regions compared to neurons. Moreover, in contrast to neurons that exhibit both intranuclear and intracytoplasmic RNA foci, glial cells show only intranuclear RNA foci [70,152]. AAV-mediated DPR expression in mice results in the accumulation of DPRs in neurons but not glial cells [66]. Further studies are needed to explain why the levels of RNA foci and DPRs are lower in glial cells. For instance, this might be due to less expression of the *C9orf72* repeated expansion mutation, less expression of RNA binding proteins involved in RNA aggregates, better *C9orf72* repeated expansion containing RNA, less RAN clearance of *C9orf72* repeated expansion containing RNA [156], less cytosolic translocation of translation, or finally, enhanced degradation of DPRs prior to their accumulation. Another possibility is that DPRs are not produced within glial cells but are instead secreted by neurons and then transferred to glial cells through possible cell-to-cell communication between neurons and glia [157]. Given the fact that glial cells undergo extensive proliferation and gliosis [158,159], this may prevent/decrease generation or dilute the level of already present RNA foci or DPRs in these cells and may provide another explanation for why levels of RNA foci or DPRs are less in glial cells compared to *C9orf72* neurons. In contrast to RNA foci and DPRs, other pathologic features of *C9orf72* repeat expansion, including TDP-43 and p62 aggregates, are present in glial cells [3,4,5,6] and even in oligodendrocytes [160,161,162] of *C9orf72* ALS/FTD cases.

#### 3.2.3. Toxic Effects of C9orf72 Glial Cells

Recent investigations have indicated that glial cells themselves may mediate neurotoxicity in *C9orf72* ALS/FTD [163]. In a study using murine embryonic stem cell-derived motor neurons, neuronal death was observed within four days of co-culture with fibroblast-derived astrocytes from *C9orf72* ALS cases [163]. Additionally, when the culture medium was partially replaced with the control astrocyte conditioned medium (ACM), the ongoing neuronal cell death was not inhibited [163]. Another study also revealed increased neuronal cell death in iPSC-derived motor neurons from either control or *C9orf72* ALS patients five days after culturing in *C9orf72* ALS ACM [164]. Moreover, it was demonstrated that induced astrocytes from *C9orf72* ALS cases can release extracellular vesicles promoting motor neuron toxicity [165]. The microRNA (miRNA) profile of extracellular vesicles secreted by *C9orf72* ALS astrocytes was found to have a unique set of 13 dysregulated miRNAs involved in axonal guidance and maintenance [165]. Among these, miR-494-3p was identified as the most dysregulated miRNA, and its reduced levels in the *C9orf72* astrocytes-secreted extracellular vesicles were correlated with dramatic consequences on axonal/neurite length and motor neuron survival in vitro and in the postmortem ALS corticospinal tract [165]. Accordingly, treatment with a miR-494-3p mimic completely rescued neurite length and the number of nodes per cell, accompanied by a significant 20–25% increase in motor neuron survival [165]. The addition of the ACM from the same *C9orf72* astrocytes also slightly affected neuronal cell survival [165]. Altogether, these studies pinpoint three important things—(i) a possible underlying gain-of-toxic-function mechanism by the *C9orf72* astrocytes in ALS/FTD pathogenesis, (ii) a level of toxicity related to possible direct physical communication between *C9orf72* astrocytes and neurons or secretion of possible neurotoxic agents from *C9orf72* astrocytes, and (iii) an impaired capacity of *C9orf72* astrocytes to support neurons. In agreement with the latter, impaired biogenesis of extracellular vesicles [47,165] or abnormal autophagy initiation [164] in *C9orf72* ALS/FTD motor neurons are linked to *C9orf72* astrocytes.

Defective adenosine triphosphate (ATP) metabolism and bioenergetic deficits are also found in the CNS of *C9orf72* ALS/FTD patients [166,167,168]. Deamination of adenosine by adenosine deaminase (ADA) generates inosine and the established pathways for nicotinamide adenine dinucleotide (NADH)-based energy production from both inosine and adenosine is through metabolism to ribose-phosphate and finally glycolysis [166,167]. A recent study found a significantly decreased ADA protein level and mRNA expression in *C9orf72* fibroblasts, induced astrocytes, and induced neurons from *C9orf72* ALS/FTD cases [166]. Notably, *C9orf72* ALS-induced astrocytes were more susceptible to adenosine-induced cell loss than control induced astrocytes, whereas inosine supplementation produced very little cell loss in any lines [166]. Additionally, it was found that defective ATP and purine metabolism due to ADA deficiency led to glial overactivation and neuroinflammation and diminished nutritional support for neurons by *C9orf72* astrocytes [166]. Loss of metabolic flexibility, involving defects in adenosine, fructose, and glycogen metabolism, and disturbances in the membrane transport of mitochondrial energy substrates are present in *C9orf72*-induced astrocytes, contributing to increased starvation-induced toxicity in these cells [167].

In addition to the above-mentioned gain-of-function mechanisms, other investigators have proposed haploinsufficiency as a possible loss-of-function mechanism underlying the glial cell-mediated toxicity in *C9orf72* ALS/FTD. One of the important actions of astrocytes is buffering the excitatory neurotransmitter glutamate in the synaptic cleft, preventing excess glutamate neurotoxicity. Notably, siRNA-mediated knockdown of both C9orf72 protein isoforms in U87 glioblastoma cells and normal human astrocytes is shown to reduce expression of excitatory amino-acid transporter (EAAT)-1 and -2, pyruvate carboxylase, and glutamine synthetase in astrocytes, and intracellular glutamate accumulation [169], providing evidence for knockdown astrocyte-related disturbed glutamate synthesis, uptake, and conversion into glutamine, which ultimately can cause glutamate excitotoxicity. These cells also exerted accumulation of p62 inclusions [169]. It was also shown that the expression of *endothelin 1*, a negative regulator of EAAT2, was increased secondary to an increased expression of nuclear factor kappa-light-chain-enhancer of activated B cells (NFκBs) due to knockdown of *C9orf72* [169]. Elevated endothelin 1 levels secreted by activated astrocytes can initiate inflammatory cascades (such as upregulation of inducible NOS and excess NO production), which cause toxicity in motor neurons [170]. Studies on *C9orf72* knockdown microglia also indicate several essential mechanisms underlying related motor neuron toxicity, including (i) elevated expression of complement component 3a receptor 1 (*C3ar1*) and complement component 1, Q subcomponent, β polypeptide (*C1qb*) in the activated *C9orf72* knockdown microglia, enhancing synaptic pruning and related demand for endosomal/lysosomal integrity [70,171], (ii) elevated expression of tyrosine kinase binding protein (*TYROBP*) and triggering receptor expressed on myeloid cells 2 (*TREM2*) in the activated *C9orf72* knockdown microglia, increasing microglial phagocytic activity [70], and (iii) a disturbed autophagy system in these microglia, causing p62 accumulation and enlarged lysosomes [70,171]. Altogether, these studies support the concept that *C9orf72* haploinsufficiency may affect glial cell function through different mechanisms that ultimately lead to motor neuron toxicity (Figure 3).

## 4. Biomarkers in *C9orf72* ALS/FTD

*C9orf72* ALS/FTD is a complex disorder linked to numerous pathologic mechanisms. Current diagnostic measures are mainly based on clinical presentation and electrodiagnostic studies [172,173], which in most cases have limitations in early diagnosis of the disease where potential treatments could be most efficacious. This signifies the importance of finding specific biomarkers that could support an early diagnosis of *C9orf72* ALS/FTD. Several studies have been conducted in recent years to identify diagnostic biomarkers for early diagnosis, disease progression, or to be used as indicators of therapeutic response in clinical trials in *C9orf72* ALS/FTD.

### 4.1. Non-Inflammatory Biomarkers

One of the specific biomarkers in *C9orf72* ALS/FTD that has caught attention in recent years is monitoring DPRs for tracking ALS in early diagnosis, natural history, and response to therapeutic intervention. Initial investigations created immunoassays for DPRs in tissue isolates (i.e., frontal/motor cortex, cerebellum, or hippocampus) from postmortem *C9orf72* ALS/FTD cases [174]. Both poly-GP and poly-GA have higher expression in the CNS of *C9orf72* ALS/FTD patients than poly-GR DPRs [175]. However, poly-GP is more likely to be correctly measured in biospecimens because it is more soluble than poly-GA [174]. Further studies have also detected poly-GP DPRs in the CSF [76,176] and peripheral blood mononuclear cells [176] from *C9orf72* ALS/FTD patients but not in healthy individuals or ALS patients who do not have the *C9orf72* repeat expansion mutation. In a more recent multi-center prospective natural history study of 116 symptomatic *C9orf72* ALS and 12 non-symptomatic *C9orf72* repeat expansion carriers, it was revealed that the CSF concentrations of poly-GP DPRs remained high and steady over time in *C9orf72* ALS patients [13]. Although there was a significant negative correlation between blood DNA repeat size and poly-GP CSF levels, no significant correlation was found between poly-GP CSF levels and ALS history measures (i.e., age at onset, survival, and ALSFRS-R rate of change) [13]. On the other hand, the stability of poly-GP levels over time may support the potential use of poly-GP as a pharmacodynamic biomarker. To investigate this, Gendron et al. (2017) evaluated the effects of a ribonuclease H-active antisense oligonucleotide (ASO) that targets G_4_C_2_ repeat RNA on lymphoblastoid cell lines from *C9orf72* ALS cases and in (G_4_C_2_)_66_ mice, and an ASO that targets intron 1 of *C9orf72* on iPSC-derived neurons from *C9orf72* ALS patients, and found that poly-GP levels were significantly decreased over time with treatment. This was concomitant with an improvement in *C9orf72* CNS pathology in mice, providing evidence that CSF poly-GP may serve as potential pharmacodynamic markers for treatments that target (G_4_C_2_)_n_ RNA [176].

The presence of nuclear RNA foci, which is another characteristic feature of *C9orf72* pathology, may be considered as another biomarker in *C9orf72* ASL/FTD. These foci, which are visualized using RNA fluorescence in-situ hybridization (FISH) techniques [8,9], are detected in both brain cells and peripheral cells, including skin biopsy-derived fibroblasts [25,33,70], lymphoblasts [70], and peripheral blood leukocytes [69], making them a potential biomarker in disease progression and in clinical trials. The detection of RNA foci in muscle biopsy tissue using the RNA-FISH technique has been broadly employed as a diagnostic marker in myotonic dystrophy type 2 [177]. This test has not yet been investigated in the muscle biopsies of *C9orf72* patients; instead, blood leukocytes may be utilized to determine whether therapies targeting *C9orf72* repeat expansion will lower the number of RNA foci in leukocytes.

### 4.2. Inflammatory/Glial Biomarkers

One of the common features of various neurodegenerative diseases including ALS is glial activation and elevated levels of inflammatory markers [133,178]. Unbiased proteomic analyses have interestingly discovered that complement activation and/or acute inflammatory responses are among the top pathways changed in the CSF of ALS patients compared to healthy individuals or other neurologic disorders [179,180]. Although detection of inflammatory mediators has been extensively investigated in ALS patients in general [181,182,183], little is known regarding their potential role as a biomarker specifically in *C9orf72* ALS/FTD cases. For instance, reduced amounts of C–X–C motif chemokine ligand 10 protein (CXCL10), a microglial chemoattractant, are found in the CSF of *C9orf72* ALS patients in comparison with other ALS cases [184]. In a recent study on FTD patients, out of the measured factors (monocyte chemoattractant protein-1 (MCP-1), regulated upon activation, normal T-cell expressed and secreted protein (RANTES), IL-10, IL-17A, IL-12p, IFN-γ, IL-8, IL-1β, leukocytes, and C-reactive protein (CRP)), only serum IL-10 was different between *C9orf72* FTD and non-*C9orf72* FTD patients (carriers had higher levels), which negatively correlated to a more rapid disease progression [185]. Clearly, more investigations are needed to identify more specific inflammatory mediators as potential biomarkers in *C9orf72* ALS/FTD.

MicroRNAs (miRNAs) are epigenetic modifiers of gene expression that act by binding argonaute 2 and forming the RNA-induced silencing complex [186]. Over 1000 miRNAs are present in humans. Alterations in miRNA expression and levels have been demonstrated in the CSF, serum, and plasma of patients with either sporadic or familial ALS [186,187,188,189,190,191]. However, there is scarce data regarding miRNAs that are specific for *C9orf72* ALS/FTD pathology compared to healthy individuals or ALS/FTD patients without the *C9orf72* repeat expansion mutation. For instance, Bengini et al. (2016) investigated the miRNA profiles of CSF from 24 ALS patients (including eight *C9orf72* ALS cases) and 24 unaffected control subjects and identified eight miRNAs as significantly deregulated in ALS (especially upregulated miR181a-5p and downregulated miR21-5p and miR15b-5p, all of which are involved in apoptotic pathways) [187]. However, no significant differences were found between ALS patients with or without the *C9orf72* repeat expansion mutation [187]. Another study also revealed that, while dysregulation of TDP-43 binding miRNAs (i.e., miR-143-5p/3p) may be a common feature of ALS pathology, downregulation of other TDP-43 binding miRNAs (i.e., miR-132-5p/3p and miR-574-5p/3p) was evident in sporadic, *TARDBP*, Fused In Sarcoma *(**FUS*) and *C9orf72*, but not superoxide dismutase 1 *(**SOD1*) mutant patients [190]. Downregulation of muscle-specific miR-206, involved in muscle re-innervation, in the *SOD1* mutant mouse accelerated the disease progression and shortened survival [191]. If miRNAs specific for *C9orf72* ALS/FTD patients are found and confirmed to be stable and secreted in the CSF (or even blood), these could be utilized as an invaluable readout for therapy efficacy. Notably, a recent study on the miRNA profile of extracellular vesicles secreted by *C9orf72* ALS astrocytes identified a unique set of 13 dysregulated miRNAs that contributed to axonal guidance and maintenance [165]. Among these, miR-494-3p was detected as the most dysregulated miRNA; its downregulation in the *C9orf72* astrocytes-secreted extracellular vesicles was correlated with dramatic consequences on axonal/neurite length and motor neuron survival in vitro and in the postmortem ALS corticospinal tract [165]. Accordingly, treatment with a miR-494-3p mimic completely rescued neurite length and the number of nodes per cell, concurrent with a robust 20–25% increase in motor neuron survival [165]. Overall, it was found that downregulated miRNAs (i.e., miR-494-3p, miR-200c-3p, miR-668-3p, and miR-140-3p) target semaphorins, RhoA, and Rock, thus, predicting an elevation in these proteins, which could result in growth cone collapse. Upregulated miRNAs (i.e., miR-297, miR-595, miR-339-5p, miR-758-3P, and miR-449a) target ephrins and WW domain containing E3 ubiquitin protein ligase 1 (Wwp1), which would cause their downregulation. Wwp1 inactivates NogoA (also called reticulon 4); thus, this could also result in axonal collapse [165]. Therefore, these miRNAs, particularly miR-494-3p, might serve as potential inflammatory markers in future studies of *C9orf72* ALS/FTD.

### 4.3. Imaging-Based Markers

Given the fact that non-invasive imaging techniques help monitor brain/spinal cord structures, neural networks, metabolism, and plasma membrane receptor distribution, they may serve as attractive markers for disease progression or treatment efficacy in ALS/FTD. Magnetic resonance imaging (MRI) and radionucleotide imaging (i.e., positron emission tomography (PET), single-photon emission computed tomography (SPECT)) are the two major techniques that have been recently investigated in this regard [192,193,194]. Consistent with previous histopathology studies on postmortem *C9orf72* ALS cases [14,195,196], recent results suggest a strong *C9orf72*-specific cortical and subcortical involvement reflecting more cognitive/behavioral deficits observed with this ALS genotype [197]. Several PET studies have also recently used radiotracers that bind to the 18 kD translocator protein (TSPO), a protein that is highly expressed on activated microglia and astrocytes, to track gliosis in ALS patients [198]. Although, in general, these studies have demonstrated that the areas of increased uptake correlated positively with upper motor neuron burden scale and negatively with ALS functional rating scale-revised (ALSFRS-R) scores [199,200,201] (two patient-reported outcome measures), little is known about whether the degree of gliosis and such correlations are different between *C9orf72* ALS/FTD and other ALS or healthy individuals. This warrants further detailed investigations.

## 5. Therapeutic Approaches: Focus on Glial Cells

Although there is still no definite cure for ALS, including *C9orf72* ALS/FTD, progress in understanding the *C9orf72* genetic architecture and its pathogenesis in ALS/FTD over the last decade has been inspiring and motivating. A better understanding of the mechanisms by which the *C9orf72* repeat expansion mutation exerts disease phenotypes has inspired multiple therapeutic approaches, providing hope for finding efficient therapies for this devastating disease in the near future. With the help of these translational approaches, several clinical trials for patients with ALS were initiated, including a phase I trial of ASOs targeting *C9orf72* variants 1 and 3 RNA (BIIB078) (clinicaltrials.gov Identifier: NCT03626012) and a phase I trial of the nucleocytoplasmic transport inhibitor KPT-350 (also called BIIB100) [202] (clinicaltrials.gov identifier: NCT03945279). Despite accumulating evidence supporting a connection between *C9orf72* ALS/FTD and neuroinflammation/autoimmunity, the fact that all trials using immunomodulatory or immunosuppressive medications (e.g., corticosteroids, cyclophosphamide, azathioprine, intravenous immunoglobulins, and plasmapheresis) have failed to show any efficacy in ALS patients [133] makes this area of research still challenging. Clearly, this issue warrants further study to understand the nature of the connection between the central inflammatory cells (i.e., glial cells) and motor neurons in the disease progress because glial cells can play both pro- and anti-inflammatory roles. Recent pre-clinical studies, for instance, have shown beneficial effects of certain *C9orf72* ALS astrocyte-related miRNAs, such as miR-494-3p, on the survival of corticospinal motor neurons from *C9orf72* ALS/FTD cases in vitro [165].

## 6. Conclusions and Perspectives

Since 2011 when the *C9orf72* repeat expansion mutation was discovered as the most common genetic abnormality in familial ALS and FTD, an impressive number of studies have markedly improved our understanding of the pathologic mechanisms underlying the *C9orf72* repeat expansion mutation. Although the normal function of *C9orf72* in humans is yet to be fully understood, either loss of function/haploinsufficiency or toxic gain of function and related downstream pathways have been suggested as underlying mechanisms involved in the pathogenesis of *C9orf72* ALS/FTD. Thus, therapies and biomarkers have been explored with respect to both mechanisms. One notable finding in recent studies is that there is an important role for glial cells in both loss- and gain-of-function theories. Here, we discussed the current literature showing pathologic roles of microglia and astrocytes in *C9orf72* ALS/FTD, including accumulating evidence of gliosis in *C9orf72* ALS/FTD, pathologic hallmarks in glial cells such as TDP-43 and p62 aggregates, and toxicity of *C9orf72* glial cells. Despite tremendous efforts on the study of glial cells in *C9orf72* ALS/FTD during the last several years, there are still several challenging issues. One important issue is that to what extent both loss- and gain-of-function theories play a concomitant role in glial cells’ involvement in *C9orf72* ALS/FTD. Data from immune system dysregulation have mainly emerged from the studies on *C9orf72* deficient animals. Although *C9orf72*^−/−^ mice exhibit severe dysregulation of the immune system and autoimmune phenotypes, more studies are clearly needed to assess a link between peripheral immune cells and CNS residual cells in disease development or progress. Concomitant modeling of loss- and gain-of-function pathways can also shed more light on the roles of microglia and astrocytes in the disease pathogenesis.

Another important issue is the cross talk between glial cells and neurons. Although limited data indicate that glial activation and neuroinflammation may precede neurodegeneration in *C9orf72* ASL/FTD, it is still elusive what factors contribute to this phenomenon, especially given the fact that pathologic *C9orf72* hallmarks of DPRs and RNA foci are less observed in glial cells than neurons, which makes the condition more challenging. Although there are several hypotheses in this regard (including less expression of the *C9orf72* repeated expansion mutation, less expression of RNA binding proteins involved in RNA aggregates, better clearance of *C9orf72* repeated expansion containing RNA, less cytosolic translocation of *C9orf72* repeated expansion containing RNA, less RAN translation, enhanced degradation of DPRs prior to their accumulation, or finally, the transference of DPRs from neurons to glial cells through possible cell-to-cell communication between neurons and glia), more studies are warranted to clearly understand this issue and test these hypotheses.

A better understanding of the underlying signaling pathways related to the aberrant function of glia in *C9orf72* ALS/FTD can provide new insights into both appropriate glial markers for disease monitoring and therapeutic approaches to slow disease progression. Thus, more studies are needed to further elucidate the roles of microglia and astrocytes in *C9orf72* ALS/FTD pathogenesis.

## Figures and Tables

**Figure 1 cells-10-00249-f001:**
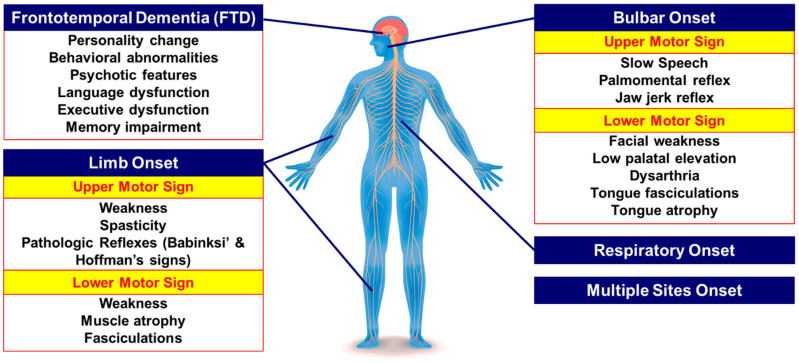
Clinical findings at onset in the chromosome 9 open reading frame 72 (*C9orf72*)-associated amyotrophic lateral sclerosis (ALS)/frontotemporal dementia (FTD).

**Figure 2 cells-10-00249-f002:**
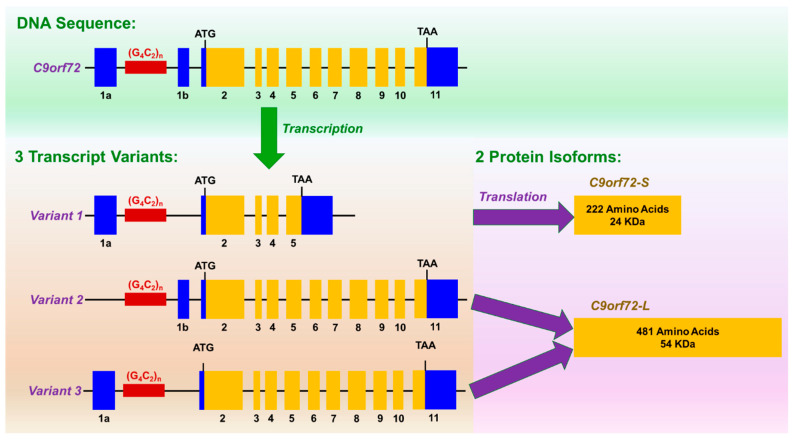
DNA sequence, three transcription variants, and two protein isoforms of *C9orf72*. The 11-exon-containing *C9orf72* gene undergoes alternative splicing, producing three transcript variants. The (G_4_C_2_)_n_ repeat expansion mutation (dark red region) is located in intron 1 of variants 1 and 3, whereas in variant 2, it is located within the promoter region. Coding exons are represented in orange and non-coding exons in blue (not to scale).

**Figure 3 cells-10-00249-f003:**
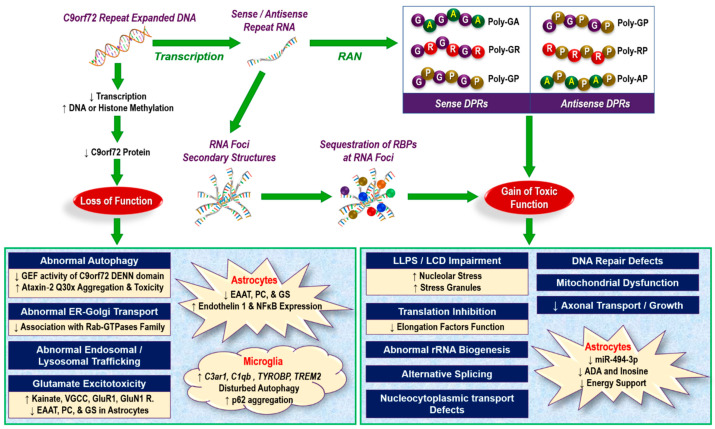
Pathogenic mechanisms implicated in *C9orf72* ALS/FTD. Both loss and gain of function mechanisms contribute to the disease process in *C9orf72* ALS/FTD. Abbreviations: ADA, adenosine deaminase; C1qb, complement component 1, Q subcomponent, β polypeptide; C3ar1, complement component 3a receptor 1; DENN, differentially expressed in normal and neoplastic cells; EAAT, excitatory amino-acid transporter; GEF, GEF, guanine nucleotide exchange factor; GluN1 R, glutamate ionotropic receptor NMDA type subunit 1; GluR1, glutamate ionotropic receptor AMPA type subunit 1; GS, glutamine synthetase; LCD, low complexity domain; LLPS, liquid–liquid phase separation; miRNA, microRNA; PC, pyruvate carboxylase; rRNA, ribosomal RNA; TREM2, triggering receptor expressed on myeloid Cells 2; TYROBP, tyrosine kinase binding protein; and VGCC, voltage-gated calcium channel.

**Figure 4 cells-10-00249-f004:**
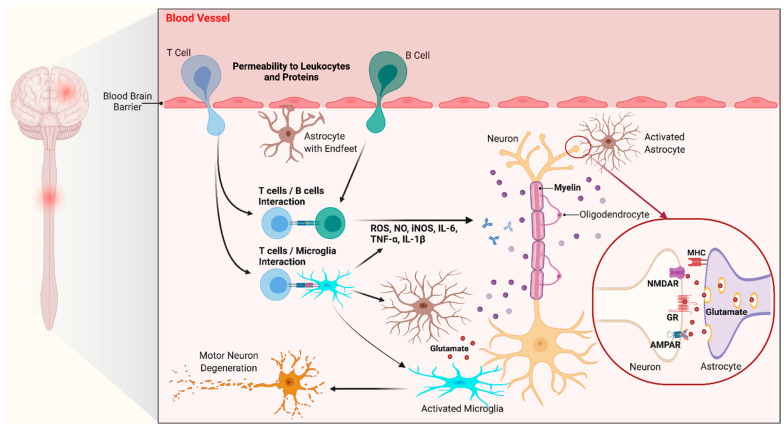
Pathogenic mechanisms underlying glial cell toxicity in neurodegeneration. Microglia and astrocytes become overactivated and lead to neurotoxicity through several mechanisms. Activated microglia and astrocytes produce excess noxious pro-inflammatory factors, such as nitric oxide (NO), reactive oxygen species (ROS, e.g., H_2_O_2_ and ONOO^−^), several cytokines (e.g., interleukin [IL]-1β, IL-6, and tumor necrosis factor [TNF]-α), and glutamate. AMPAR, α-amino-3-hydroxy-5-methyl-4-isoxazolepropionic acid (AMPA) receptor; GR, glutamate receptor; MHC, major histocompatibility complex; and NMDAR, *N*-methyl-D-aspartate (NMDA) receptor.

**Table 1 cells-10-00249-t001:** Immune system dysregulation phenotypes in homozygotes in mouse models of *C9orf72* loss of function.

*C9orf72* Knockout Method	Immune System	Motor	Cognition/Behavior	Ref
Non-conditional exons 2–6	Splenomegaly and cervical lymphadenopathy	Mild motor deficits only on a rotarod assay at 12 months	Mild social interaction and social recognition deficits	[58]
Non-conditional full gene	Splenomegaly, systemic lymphadenopathy, glumerulonephropathy, and ↑ serum IL-12, IL-17a, IL-10, TNF-α, plasma cells, and activated T cells.	Mild motor deficits, tremor, and rigidity at 40 weeks	NR	[59]
Non-conditional exons 2–6 or zinc finger deletion	Splenomegaly and cervical lymphadenopathy	Normal function	NR	[60]
NR, all tissues full knockout	Splenomegaly, cervical lymphadenopathy, and B-cell lymphomas	Normal function	NR	[46]
Non-conditional exons 2–6 knockouts in a C57BL/6 background or CRISPR/Cas9	Splenomegaly, cervical lymphadenopathy, hepatomegaly, and ↑ serum IL-22, IL-28, IL-23, IL-6, MCP-1, IL-31, IL-5, IL-10, IL-1β, IL-15/IL-15R, IFNγ, IL-3, GM-CSF, IL-17A, IFNα, MIP-1B, LIF, GROα	NR	NR	[61]
Non-conditional CRISPR/Cas9	Splenomegaly and systemic lymphadenopathy	NR	NR	[63]
Non-conditional exons 2–6	Splenomegaly	NR	Lethargy	[62]
Non-conditional exons 2–6	Splenomegaly, systemic lymphadenopathy, and ↑ serum IL-6	NR	NR	[134]
Non-conditional exons 2–6	Splenomegaly	Normal function at 3 months	NR	[135]

GM-CSF, granulocyte-macrophage colony-stimulating factor; GROα, growth-regulated oncogene α; IL, interleukin; LIF, leukemia inhibitory factor; MIP, macrophage inflammatory protein; NR, not reported; TNF, tumor necrosis factor; and ↑, increased.

**Table 2 cells-10-00249-t002:** Evidence of gliosis in *C9orf72* ALS/FTD.

Species	Region	Results	Ref
*C9orf72^−/−^* mice	Brain	↑ LysoTracker- and Lamp1-positive structures in microglia	[60]
	Isolated spinal cord microglia	↑ IL-6 and IL-1b levels	
*C9orf72^−/−^* mice	Brain and spinal cord	No change in GFAP and Iba1 staining at 18 months	[57]
(G_4_C_2_)_500_ BAC transgenic mice	Hippocampus	↑ Iba1 staining in acute end-stage (20–40 weeks) mice	[139]
	Motor cortex (layers I–III & layer V) and hippocampus	↑ GFAP staining in acute end-stage (20–40 weeks) mice	
Transgenic mice expressing poly-(GA)_149_	Spinal cord	↑ CD68 and Iba1 immunostaining and mRNA expression at six months, but little at one month; No change in GFAP immunostaining and mRNA expression	[100]
Transgenic mice expressing poly-(GA)_50_	Brain	↑ GFAP mRNA expression, but No change in Iba1 mRNA expression at six months	[99]
	Cortex, motor cortex, and hippocampus	↑ GFAP immunostaining and immunohistochemistry	
Transgenic mice expressing poly-(GR)_100_	Brain	↑ GFAP and Iba1 mRNA expression and immunostaining at 1.5 > 3 > 6 months	[114]
*C9orf72* ALS patients	Postmortem motor cortex and spinal cord	↑ Iba1 and Lamp1 immunostaining	[60]
*C9orf72* ALS patients	Postmortem pyramidal tract at all levels (white matter underlying motor cortex, mid-crus cerebri, medullary pyramids, and lateral and anterior corticospinal tracts)	↑ CD68 immunohistochemistry	[4]
*C9orf72* ALS patients	Postmortem white matter of the medulla and the motor cortex	↑ CD68 and Iba1 immunostaining	[141]
*C9orf72* ALS patients	Postmortem corpus callosum	↑ CD68 immunohistochemistry in the body more than genu or splenium of the callosum	[142]
*C9orf72* FTD patients	Plasma	No change in GFAP concentration between pre- and symptomatic cases and non-carriers	[143]
*C9orf72* ALS patients	[^18^F]FDG PET in *C9orf72* ALS vs. sporadic ALS	↓ Metabolism in the anterior and posterior cingulate cortex, insula, caudate and thalamus, the left frontal and superior temporal cortex, ↑ Metabolism in the midbrain, bilateral occipital cortex, globus pallidus, and left inferior temporal cortex	[140]

ALS, amyotrophic lateral sclerosis; FTD, frontotempoal dementia; GFAP, glial fibrillary acidic protein; Iba1, Ionized calcium-binding adaptor molecule 1; IL, interleukin; ↑, increased; and ↓, decreased.

## Data Availability

Not applicable.

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
