# Peer review of "Glial Cell Dysfunction in C9orf72-Related Amyotrophic Lateral Sclerosis and Frontotemporal Dementia"

_cells, 2021, doi:10.3390/cells10020249_

Round 1

Reviewer 1 Report

This review article from Dr. Ghasemi and colleagues is a well written and thoroughly researched description of the state of glia cells’ role in C9ORF72 ALS.  Few comments are listed below.

  1. Figure legends tend to simply repeat the main text, particularly for Figure 1 and 2. I would suggest shortening the figure legends or revising their content.
  2. Page 3, lines 83-85. This sentence is written in a potentially misleading way as it seems to suggest that V2 levels are higher than V1, which is not the case. It is also unclear if the brain regions mentioned refer to fetal or adult tissue. I would suggest revise the sentence such as follows: “Compared to variants 1 and 3, expression of variant 2 is higher in the central nervous 83 system (CNS) relative to other tissues [27,28] especially in the fetal brain and adult cerebellum and frontal cortex, and with lower expression in the hippocampus [8].”
  3. Page 3, line 85-87. The reference cited here refers to a transcriptomic analysis of C9orf72 ALS and has no immunohistochemistry data. More appropriate ref would be Frick et al, 2018. The authors may also want to mention the possible distinct localization of the long and short C9ORF72 isoforms (Xiao et al., 2015) as well as the potential developmental changes in its cellular localization (Atkinson et al., 2015).
  4. Page 5, line 169. GuR1 should be GluR1
  5. Page 7, Line 251. ploy-GA should be poly-GA

Author Response

Response to Reviewer 1 Comments

This review article from Dr. Ghasemi and colleagues is a well written and thoroughly researched description of the state of glia cells’ role in C9ORF72 ALS.  Few comments are listed below.

Point 1: Figure legends tend to simply repeat the main text, particularly for Figure 1 and 2. I would suggest shortening the figure legends or revising their content.

Response 1: As the reviewer recommended, figure legends for figure 1, 2 and 4 were shortened significantly. The changes are on as follows:

  • Figure legend 1 on page 2.
  • Figure legend 2 on page 3.
  • Figure legend 4 on page 8.

Point 2: Page 3, lines 83-85. This sentence is written in a potentially misleading way as it seems to suggest that V2 levels are higher than V1, which is not the case. It is also unclear if the brain regions mentioned refer to fetal or adult tissue. I would suggest revise the sentence such as follows: “Compared to variants 1 and 3, expression of variant 2 is higher in the central nervous 83 system (CNS) relative to other tissues [27,28] especially in the fetal brain and adult cerebellum and frontal cortex, and with lower expression in the hippocampus [8].”

Response 2: We thank the reviewer for this note. This was corrected according to the reviewer recommendation on page 3, line 81-83.

Point 3: Page 3, line 85-87. The reference cited here refers to a transcriptomic analysis of C9orf72 ALS and has no immunohistochemistry data. More appropriate ref would be Frick et al, 2018. The authors may also want to mention the possible distinct localization of the long and short C9ORF72 isoforms (Xiao et al., 2015) as well as the potential developmental changes in its cellular localization (Atkinson et al., 2015).

Response 3: We thank the reviewer very much for this deep point of view. As the reviewer recommended, the reference “Frick et al. (2018)” was added as more appropriate reference for immunohistochemistry data and neuronal cytoplasmic protein, localizing largely at the presynaptic terminals (page … line). Additionally, we added more descriptions for distinct localization of the long and short C9orf72 isoforms as well as developmental changes in its cellular localization on page 3, line 83-91.

Point 4: Page 5, line 169. GuR1 should be GluR1

Response 4: “GuR1” was replaced with “GluR1” on page 5, line 171.

Point 5: Page 7, Line 251. ploy-GA should be poly-GA

Response 5: “ploy-GA” was replaced with “poly-GA” on page 7, line 273.

Reviewer 2 Report

In this review the Authors summarized the data on the pathogenesis of C9orf72-related ALS and FTD. The manuscript is well written, although slightly unbalanced. Indeed, despite it should be focused on glial dysfunction, a large part of the paper is dedicated to a broader description of the pathogenetic mechanisms involved in C9orf72 RE ALS-FTD.

Since the role of glial cells in C9orf72 RE is the main focus of the review, it could be useful for the reader to have a table summarizing the main findings (and eventually also the missing evidences) supporting  the involvement of these cells possibly differentiating it for glial cell (i.e. astrocytes vs microglia related observations) and source of evidence (i.e. in vitro, animal models, human data).  

It could be useful for the not expert reader to clarify the significance of the expression of certain microglial markers (i.e. CD68) also in relation to the quiescent/activate phenotype. Also, it could be useful to explain that an increased CD68 expression does not necessary imply an M1 phenotype (in paragraph 3.2.1).

In the 4.2 paragraph, it could be useful to integrate some information about other markers of glial activation such as GFAP, sTREM2 for instance.

The conclusion and perspective paragraph should be expanded and should indicate what are the missing pieces to understand the role (either positive or detrimental) and relevance of glial cells in C9orf72 pathology, suggesting also possible useful approaches to achieve this goal.

Minor points

Introduction - Although cognitive impairment has been found in up to about 50% of patients with ALS, more frequently it has been shown in 30% of cases, therefore it would be more appropriate to indicate a range of 30-50 instead of about 50%.

Line 283; this sentence doesn’t seem correct; did the author mean highly evolved?

Line 311, Endfeet shouldn’t start with a capital letter.

Iba1 is misspelled as ib1a several time; please revise.

Author Response

Response to Reviewer 2 Comments

Point 1: In this review the Authors summarized the data on the pathogenesis of C9orf72-related ALS and FTD. The manuscript is well written, although slightly unbalanced. Indeed, despite it should be focused on glial dysfunction, a large part of the paper is dedicated to a broader description of the pathogenetic mechanisms involved in C9orf72 RE ALS-FTD.

Since the role of glial cells in C9orf72 RE is the main focus of the review, it could be useful for the reader to have a table summarizing the main findings (and eventually also the missing evidences) supporting the involvement of these cells possibly differentiating it for glial cell (i.e. astrocytes vs microglia related observations) and source of evidence (i.e. in vitro, animal models, human data).

Response 1: We thank the reviewer for his/her important comment. Accordingly, we added two tables to the revised manuscript. Table 1 represents the immune system dysregulation phenotype in mouse loss of C9orf72 function. In Table 2, we described the current evidence for gliosis (either astrogliosis or microgliosis) in previous studies on C9orf72 animal models as well as C9orf72 ALS/FTD cases.

  • Table 1: Page 9.
  • Table 2: Page 10.

Point 2: It could be useful for the not expert reader to clarify the significance of the expression of certain microglial markers (i.e. CD68) also in relation to the quiescent/activate phenotype. Also, it could be useful to explain that an increased CD68 expression does not necessary imply an M1 phenotype (in paragraph 3.2.1).

Response 2: As the reviewer recommended, we had more clarification on activated microglial markers including CD68 and Iba1. In agreement with the reviewer, we also mentioned that CD68 is not only expressed on activated microglia and it can also be expressed on resting microglia. The changes are on page 11, line 376-382.

Point 3: The conclusion and perspective paragraph should be expanded and should indicate what are the missing pieces to understand the role (either positive or detrimental) and relevance of glial cells in C9orf72 pathology, suggesting also possible useful approaches to achieve this goal.

Response 3: We thank the reviewer for this comment. As the reviewer recommended, we expanded the conclusion and perspective session on page 16, line 659-681.

Minor Points

Point 4: Introduction - Although cognitive impairment has been found in up to about 50% of patients with ALS, more frequently it has been shown in 30% of cases, therefore it would be more appropriate to indicate a range of 30-50 instead of about 50%.

Response 4: As the reviewer recommended, 50% was changed to 30-50% to have more accurate range for frequency of cognitive decline in ALS patients. This change is on page 1, line 29.

Point 5: Line 283; this sentence doesn’t seem correct; did the author mean highly evolved?

Response 5: This was a typo; thus, “highly involved” was replaced with “highly evolved” on page 7, line 285.

Point 6: Line 311, Endfeet shouldn’t start with a capital letter.

Response 6: “Endfeet” was replaced with “endfeet” on page 8, line 313.

Point 7: Iba1 is misspelled as ib1a several time; please revise.

Response 7: We do apologize for this typo. This was corrected throughout the manuscript on page 11, second paragraph (lines 388, 394, 397, 398, and 400).

Round 2

Reviewer 2 Report

The manuscript has definitively improved. I do not have further comments.